# A Comprehensive Review of the Evolution of Insulin Development and Its Delivery Method

**DOI:** 10.3390/pharmaceutics14071406

**Published:** 2022-07-04

**Authors:** Vaisnevee Sugumar, Kuan Ping Ang, Ahmed F. Alshanon, Gautam Sethi, Phelim Voon Chen Yong, Chung Yeng Looi, Won Fen Wong

**Affiliations:** 1School of Medicine, Faculty of Health & Medical Sciences, Taylor’s University, 1, Jalan Taylors, Subang Jaya 47500, Malaysia; vaisneveesugumar@sd.taylors.edu.my; 2Department of Medical Microbiology, University Malaya Medical Center, Kuala Lumpur 59100, Malaysia; angkp@ummc.edu.my; 3Center of Biotechnology Researches, University of Al-Nahrain, Baghdad 10072, Iraq; ahmed.neama@brc.nahrainuniv.edu.iq; 4Department of Pharmacology, Yong Loo Lin School of Medicine, National University of Singapore, Singapore 117600, Singapore; 5School of Biosciences, Faculty of Health & Medical Sciences, Taylor’s University, 1, Jalan Taylors, Subang Jaya 47500, Malaysia; phelim.yong@taylors.edu.my; 6Centre for Drug Discovery and Molecular Pharmacology (CDDMP), Faculty of Health & Medical Sciences, Taylor’s University, 1, Jalan Taylors, Subang Jaya 47500, Malaysia; 7Department of Medical Microbiology, Faculty of Medicine, University of Malaya, Kuala Lumpur 50603, Malaysia

**Keywords:** diabetes mellitus, transdermal, chemical enhancers, physical enhancers, non-invasive insulin delivery

## Abstract

The year 2021 marks the 100th anniversary of the momentous discovery of insulin. Through years of research and discovery, insulin has evolved from poorly defined crude extracts of animal pancreas to recombinant human insulin and analogues that can be prescribed and administered with high accuracy and efficacy. However, there are still many challenges ahead in clinical settings, particularly with respect to maintaining optimal glycemic control whilst minimizing the treatment-related side effects of hypoglycemia and weight gain. In this review, the chronology of the development of rapid-acting, short-acting, intermediate-acting, and long-acting insulin analogues, as well as mixtures and concentrated formulations that offer the potential to meet this challenge, are summarized. In addition, we also summarize the latest advancements in insulin delivery methods, along with advancement to clinical trials. This review provides insights on the development of insulin treatment for diabetes mellitus that may be useful for clinicians in meeting the needs of their individual patients. However, it is important to note that as of now, none of the new technologies mentioned have superseded the existing method of subcutaneous administration of insulin.

## 1. Introduction

Etymologically, the term “diabetes mellitus” is taken from both Greek and Latin words. “Diabetes” in ancient Greek means siphon or to pass through, whereas the Latin word “mellitus” means sweet taste. The ancient Greeks used to diagnose the disease by directly tasting a patient’s urine. It was not until the 19th century that a clinical test was developed to test for diabetes. This test was invented by Karl Trommer and tested for sugar in the urine using acid hydrolysis. Around 1894, Sir Edward Albert Sharpey-Schäfer suggested that the pancreatic islets might drive the effects of the pancreas on blood sugar control. Although he did not isolate the insulin protein, he coined the term “insulin” to describe this yet undiscovered substance. The discovery of insulin occurred in 1921 following the ideas of a Canadian orthopedic surgeon, Frederick G. Banting; the chemistry skills of his assistant, Charles Best; and John MacLeod of the University of Toronto. The discovery of insulin was a watershed moment when DM changed from a certain death sentence to a manageable disease. This marks a new chapter for DM management, as before 1921, it was exceptional for people with type 1 diabetes to live more than a year or two [1].

## 2. The Pathogenic Landscape of Diabetes Mellitus (DM)

Although 2021 marks the 100th anniversary of insulin’s discovery, the pathogenesis of DM remained unclear until recent decades. Following years of research, we have come to know that type 1 diabetes mellitus (T1DM) is associated with the destruction and dysfunction of pancreatic β-cells, causing a shortage of systemic insulin supply. On the other hand, type 2 diabetes mellitus presents as insulin resistance, where the cells are desensitized from insulin-related signal transduction and mitochondrial exhaustion. Both type 1 and type 2 diabetes lead to the accumulation of non-metabolized blood sugar, which contributes to the advancement of diabetic complications.

Autoimmunity initiates the pathogenesis of type 1 diabetes mellitus (T1DM). Recent polychromatic flow cytometry results showed that natural killer (NK) cells, dendritic cells (DC), and T lymphocytes are the cardinal elements of T1DM pathogenesis [2]. The autoreactive T-lymphocyte infiltration of the pancreatic islet is the result of central and peripheral tolerance impairments [3], the propagation of which can be influenced by the existence of native antigen post-translational modification [4] and the viral antigen [5,6,7]. The roles of dendritic cells in the destruction of pancreatic islets has been illustrated and proven previously, as dendritic cells are crucial for antigen presentation with respect to cellular immunity [8,9,10]. In addition, merocytic dendritic cells, a type of Zbtb46-regulated, IRF4-activated dendritic cell, exhibited a promoting role in the murine type 1 diabetes mellitus model [11,12,13,14]. Furthermore, the roles of NK cells T1DM pathogenesis were illustrated previously [15]. However, the overall mechanism of NK cells remains elusive. 

In contrast, deficits of insulin secretion and insulin intolerance play cardinal roles in the pathogenesis of T2DM [16]. One of the factors that causes a deficit in insulin secretion is obesity, whereby ectopic lipid accumulation leads to excessive oxidative stress on the endoplasmic reticulum and triggers apoptosis of pancreatic β-islet cells [17]. Additionally, lipid accumulation causes glucolipotoxicity, impairs cell autophagy, and causes pancreatic β-islet cell apoptosis [18]. In addition, dyslipidemia affects mitochondrial function via oxidative stress, activates serine/threonine-kinase-related pathways, and increases the phosphorylated serine of IRS protein, contributing to the progression of insulin intolerance [19,20]. Nevertheless, genetic predisposition [21] and the host microbiome [22] also contribute to the progression of T2DM. 

## 3. The Evolution of Insulin Development

Insulin was discovered by Sir Frederick G. Banting, Charles H. Best, and J.J.R. Macleod at the University of Toronto in 1921 (Figure 1), and it was subsequently purified by James B. Collip. In January 1922, Leonard Thompson, a 14-year-old boy, received his first injection of animal insulin and survived the complications of diabetes mellitus [1,23]. Through a series of protein chemistry and x-ray crystallographic studies, the biochemistry of insulin and its corresponding receptor was elucidated. In general, insulin is a heterodimeric peptide that consists of two chains, A chain (21 amino acids) and B chain (30 amino acids), which appear in a two-Zn2+-stabilized hexamer [24]. The two chains join together via two interpeptide disulfide bonds (CysB7 to CysA7 and CysB19 to CysA20) and an intrapeptide disulfide bond (CysA6 to CysA11) [25]. Through the interaction between insulin and IGF1-receptor binding, the receptor’s tyrosine is phosphorylated and cascaded to the downstream signal pathways involved in lipid kinase, as well as PI3K- and AKT-related pathways [26]. 

Previously, patients were required to receive multiple insulin injections in a day due to the short duration of the action of insulin; however, in 1936, scientists attempted to prolong the duration of insulin action. H.C. Hagedorn of Denmark developed protamine-enhanced insulin hexamer to prolong the release of insulin and its activity, whereas Scott and Fisher created a zinc-enhanced lente insulin for a similar reason [27,28]. These two approaches were combined to create insulin with a longer duration of activity (at least 24 h), namely neutral protamine Hagedorn (NPH) insulin. NPH insulin was manufactured and marketed by Nordisk in 1950 [27,28]. It serves as intermediate-acting insulin, employing a release delay mechanism of protamine [29]. Although the insulin activity was prolonged, the source of insulin at that time was of bovine or porcine origin, whereas protamine originated from salmon sperm. These animal sources trigger the immune response once administered, causing progression to hypersensitivity [30]. Hence, human semisynthetic insulin was created in 1978 through the expression of recombinant bacterial plasmid after a half century of animal-based insulin application. Recombinant human insulin was synthesized and marketed in 1983 by Eli Lilly and Company and named Humulin [31,32,33]. Although the risk of hypersensitivity is reduced following the use of human native insulin, its application is limited by variable peaks and duration of action onset. Furthermore, human native insulin contributes to the risk of hypoglycemia and poor pharmacokinetic profiles with unpredictable absorption rates and action. 

Therefore, insulin analogs (basal and prandial types) were developed via genetic engineering and systemic biological approaches [34,35]. Prandial insulin analogs resolve the limitations of native human insulin through faster absorption and action onset (within 15 min before meals) to compensate for the spike in blood sugar after meals and the high insulin demand during diabetic complications. Lispro (branded as Humalog) is the first prandial, fast-acting insulin analog with modifications of B28 lysine and B29 proline and was approved for market in 1996. Such modifications minimize the self-association of insulin molecules and allow the insulin to appear in active monomeric form [36,37]. Lispro works in the body within 15 min after administration. Aspart (branded as Novolog and Novorapid) a short-acting prandial insulin analog approved and marketed in 2000 [28]. It resulted in the modification of aspartic acid in the B28 position [38]. Aspart reacts faster than lispro, as aspart works within 10 min. However, the two sugar control outcomes are indistinguishable in type 1 diabetes mellitus [39]. Glulisine is another newer, short-acting prandial insulin analog compared to lispro and aspart and was approved and marketed since 2004 [28]. Glulisine was modified via B3 lysine substitution and B29 glutamate substitution [40], resulting in the same sugar control outcomes as those observed with lispro and aspart in type 1 diabetes mellitus [41]. A systematic review and meta-analysis comparing prandial insulin analogs and native human insulin with respect to sugar control of type 1 diabetes mellitus revealed that prandial insulin analogs are superior to native human insulin in terms of episodes of total hypoglycemic events, postprandial glucose, and HbA1c outcomes [42,43]. However, in a type 2 diabetes mellitus study group, there was no distinctive difference [44] or the results were inconclusive [43] in terms of beneficial outcomes. 

On the other hand, basal insulin analogs work as long-acting insulin to provide a constant, minimal insulin secretion level and stabilize the blood sugar level between meals and overnight. A previous study on type 1 diabetes mellitus showed that basal insulin analogs are superior to NPH with respect to risk reduction in nocturnal and severe hypoglycemia [45]. Similar findings were observed in type 2 diabetes [46,47]. As for the types of basal insulin analogs, glargine (branded as Lantus) was the first basal insulin analog marketed in 2000. It was developed via the insulin modification of two additional arginine molecules at the C-terminus of the B chain and A21 glycine modification [48]. Such modifications altered its solubility in the subcutaneous layer via the change in pH, as an alteration in pH causes higher-order aggregation and controls the release of insulin into the blood stream for at least 24 h [28,49]. Besides glargine, detemir (branded as Levemir) is another long-acting insulin that can bind to albumin as a result of B29 lysine—myristic acid modification. Such modification prolongs its activity through slow release from the albumin [50,51]. Detemir was approved for the market in June 2005 [28]. In addition, a previous study was conducted to compare both efficacies in the type 2 diabetes management. The result showed that once-daily injection of glargine was required to achieve glycemic control, whereas a twice-daily injection of detemir at a higher dose was required to achieve similar results [52]. The results in type 1 diabetes mellitus showed that glargine is not inferior to detemir in terms of glycemic control [53,54]. 

In 2012, a novel ultra-long-acting insulin analog known as degludec (branded as Tresiba) [55]. The degludec analog is insulin with two alterations: (1) the removal of B chain threonine in position 30 and (2) the attachment of 16-carbon fatty diacid (hexadecenoic diacid) on the B-chain lysine in position 29 through glutamate acylation. These modifications permit the formation of multi-hexamers and appear as a depot complex in the subcutaneous layer, leading to the slow release of insulin into systemic circulation [55,56]. Regarding the glycemic control efficiency of these three basal insulin analogs, a meta-analysis was conducted to compare glargine, detemir, and degludec with respect to their glycemic control in type 1 and type 2 diabetes. The overall outcomes showed no differences among three [57]. However, degludec was found to be less susceptible to episodes of hypoglycemia compared to glargine and detemir [57]. Lastly, icodec is a novel insulin analog administered weekly via the oral route with a plasma half-life of 196 h [58]. Through amino acid substitutions (A14E, B16H, and B25H) and the addition of 1,20-icosanedioic acid (C20), these warrant a strong and reversible albumin binding activity, together with a stable and longer half-life molecular structure [58,59].

## 4. Improvement in Pharmacokinetic/Pharmacodynamic (PK/PD) Profiles

Regular human insulin (RHI) is a short-acting insulin that takes 30 min to 1 h for action onset, 2 to 4 h for peak activity, and 6 to 8 h to sustain the effect [28,60] (Table 1). Attributable to the short-acting nature of human insulin, neutral protamine Hagedorn (NPH) was created to serve as an intermediate-acting insulin in 1946 (Holleman et al., 2007). NPH is a combination of protamine and animal insulin in the “isophane” form, wherein the insulin is dissolved in a zinc-contained buffer to stabilize the mixture at neutral pH [29,34]. In the context of pharmacokinetics, NPH takes 1–3 h to take effect, 6–10 h to achieve peak activity, and 14–24 h to maintain the drug effect [29] (Table 1). However, these human native insulin products share the risk of late postprandial hypoglycemia and variable glucose-reducing profiles, which can be a major barrier for effective diabetes management [52]. Thus, the aim of the advancement of insulin analogs is to overcome these issues. Insulin analogs were developed via amino acid substitution and modifications to achieve the following goals: mimicry of physiological insulin secretion, adequate management of fasting and postprandial blood sugar profiles, and minimal hypoglycemic risk [1]. These analogs have been formulated into either bolus insulin analogs (lispro, aspart, and glulisine) or basal insulin analogs (glargine, detemir, degludec, and icodec) (Table 2).

### 4.1. Bolus Insulin Analogs

The first bolus insulin analog applied clinically was lispro. It was designed by substituting amino acid B28 proline with lysine and B29 lysine with proline in the B chain of the insulin molecule. This prevents self-association of insulin dimer/hexamer and allows more insulin monomers to distribute in the subcutaneous layer, permitting a rapid sugar-lowering effect after administration (DrugBank accession number: DB00046) [61,62]. Lispro begins to respond within 5 to 15 min after subcutaneous injection and achieves peak serum concentration (Cmax) within 30–60 min (DrugBank accession number: DB00046) [61] (Table 1). The potency of one mole of lispro molecules is equivalent to one mole of native human insulin [63]. Lispro can maintain its drug of action for 3–4 h before receptor-mediated degradation. The absolute bioavailability of lispro can reach up to 77% with doses between 0.1 and 0.2 units/kg. 

Aspart has a comparable profile to that of lispro, including its biopotency [64]. Aspart is formed via substitution of amino acid B28 proline for aspartic acid. This modification eliminates the monomer–monomer attractive surface and further enhances the repulsion between the charged aspartic acid and nearby amino acid [65]. As a result, the insulin hexamer rapidly dissociates into a monomer within 10–20 min after subcutaneous administration (DrugBank accession number: DB01306; Novo Nordisk, Bagsværd, Denmark). It achieves peak serum concentration after 40–50 min post onset with a mean maximum concentration (Cmax) of 82 mU/L (Table 1), and the drug action is sustainable for 3–5 h [38]. 

Glulisine is a newer type of bolus insulin analog (DrugBank accession number: DB01309) [66]. It is made up of two amino acid substitutions. First, asparagine in the B3 position is replaced by lysine. Secondly, lysine in the amino acid position B29 is replaced by glutamic acid. These modifications decrease the isoelectric point of insulin from 5.5 (native insulin) to 5.1, which consequently improves insulin solubility in the subcutaneous layer [65,67]. Glulisine has a rapid onset of action within 20 min after subcutaneous administration and then achieves a Cmax of 83 microunits/mL one hour post administration (DrugBank accession number: DB01309) [67] (Table 1). Glulisine can sustain the drug of action for 4 h [67], and its absolute bioavailability is approximately 70% following administration (DrugBank accession number: DB01309).

### 4.2. Basal Insulin Analogs

Basal insulin analogs achieve a longer interval of action via fixed and maximal insulin level for hours (known as a peakless profile) [68]. Therefore, they can serve as basal insulin replacements. Glargine is the first basal insulin analog, and it was introduced into the market in 2000. Glargine has two L-arginine residues added to the C terminal of the B chain and an amino acid substitution of glycine for asparagine. These modifications change the isoelectric point of insulin and prevent the deamidation effect of asparagine, subsequently producing more stable insulin aggregation for long-term release [48] (Table 2). Glargine starts to react within 1–2 h after administration. It can sustain the drug action for approximately 24 h, with a Cmax of 18.9 mU/mL (DrugBank accession number: DB00047; Sanofi-Aventis US) (Table 1). Detemir is another basal insulin analog. It was produced in 2004 by Novo Nordisk through the acylation of myristic acid to the lysine of B-chain amino acid position 29 [69]. This modification allows detemir to reversibly bind albumin via myristic acid and slow down the insulin release, prolonging the sugar-lowering duration [70] (Table 2). Onset occurs after 1.6 h onset, with a Cmax of 149 pmol/L (DrugBank accession number: DB01307) (Table 1), and the standard dosage can sustain drug action for up to 24 h [70,71].

Degludec is another novel type of basal insulin analog with ultra-long duration [72]. Degludec has the same amino acid sequence as human insulin, except for the removal of threonine in position 30 of the B chain and the attachment of lysine in position 29 of the B chain via a glutamic acid linker of a 16-carbon fatty diacid. These alterations promote insulin depot formation in the subcutaneous layer through multi-hexamer aggregation, whereby they dissociate according to a predictable gradient and release the insulin monomer into the blood stream for a longer duration [73]. Cmax of degludec can reach 4472 pmol/L within a one-hour onset time with a subcutaneous dosage of 0.4 U/kg, (DrugBank accession number: DB09564). The drug action can be maintained for at least 42 h and to a maximum of 4 days [74] (Table 1). 

Lastly, icodec is a revolutionary basal insulin analog, which was recently developed in 2020/2021 [58]. Icodec was created with two modifications: the addition of C20 fatty diacid to the B-chain amino acid position 29 lysine, together with the deletion of B-chain amino acid position 30 threonine, allowing for effective and reversible albumin binding [58]. Secondly, there are three amino acid substitutions (A14E, B16H, and B25H), which enable slower receptor-mediated clearance and prolong the half-life of the analog [59](Table 2). Currently, icodec is undergoing a phase III trial [75]. According to a report, icodec can sustain for one week with a maximum plasma concentration of 16 h, and this analog has a maximum half-life of 196 h [76] (Table 1).

## 5. Advancements in Insulin Delivery Technology

Despite the advancements in oral drug therapy for the management of diabetes mellitus, the mainstay treatment remains injectable insulin. This is because drugs administered orally result in low bioavailability, mainly due to the first-pass metabolism effect of the body. Degradation by gastrointestinal enzymes is likely through this route [77,78]. Hence, the alternative treatment is to administer insulin parenterally. However, there are several limitations associated with this route of delivery. First, the constant requirement for insulin injection leads lower compliance among patients due to pain at the site of injection and discomfort after administration [79]. The ultimate goal of subcutaneous insulin therapy is to mimic normal physiological insulin in order to attain normoglycemia. Nevertheless, this is not always successful, largely due to the altered absorption of insulin [80]. Furthermore, one of the most common complications associated with frequent injections is lipohypertrophy, whereby fat tissue accumulation occurs on the normal skin surface, substantially delaying the absorption, as well as the bioavailability, of insulin [81,82]. However, various efforts have been employed to address these issues, such as the use of chemical and physical enhancers through different routes of administration (Figure 2) for the delivery of insulin. 

### 5.1. Chemical Enhancers

#### 5.1.1. Nasal

Intranasal delivery has long been investigated as a possible route of administration for insulin. This route is advantageous, mainly due to its non-invasive method, which is comparable to that of the parenterally administered insulin. Additionally, the intranasal route could result in an increase in the bioavailability and absorbability of insulin into the systemic circulation through the epithelial membrane. This would indirectly prevent the degradation of insulin by the enzymatic activity in the gastrointestinal tract [83,84]. Multiple studies have highlighted the potential use of carriers and enhancers for the intranasal delivery of insulin.
In Vitro

Despite being widely used as a drug carrier, chitosan on its own would not be able to deliver insulin across the nasal membrane due to its large molecular size, making the use of enhancers along with chitosan necessary for the delivery of insulin. Hence, Bahmanpour et al. formulated a combination of chitosan (Ch) and quaternized chitosan ammonium salt (HTCC) to investigate its action as a permeation enhancer. Additionally, gelatin (Gel) was added into the formulation to increase the adhesiveness and absorption of the drug into the nasal membrane. It was observed that the formulation with 1:6 ratio of gel to Ch-HTCC resulted in a slower release of insulin: 65% in 10 h as compared to 70% within 1 h when the ratio of Ch-HTCC was reduced to 3. The authors concluded that gel: Ch-HTCC (1:6) consisting of 2 wt% chitosan, 1 wt% HTCC, and 0.5 wt% Gel provides the best results with a prolonged release of insulin rate. This formulation could be used in the future for delivery of insulin in a controlled release and for a prolonged time [85].
In Vivo

Cell-penetrating peptides (CPPs) are a category of peptides that are capable in enhancing the permeability of large therapeutic proteins and peptides across cellular membranes [86]. The effect of various CPPs on the nasal absorption of insulin was evaluated in an in vivo rat model. It was observed that administration of insulin, along with tested CPPs (+L-R8, +D-R8, +D-penetratin, and +L-penetratin) showed an enhanced absorption into the nasal mucosa. Specifically, the reduction in blood glucose after the coadministration of +D-penetratin and +L-penetratin with insulin was 30% and 50%, respectively, as compared to R8. Subsequently, the dose-dependent effect of D- and L-penetratin on nasal insulin bioavailability was evaluated. The increase in concentration of L-penetratin from 0.2 mM to 2 mM increased the permeability of insulin into the nasal mucosa by 8.8 to 30 times, which was more than that of insulin solution alone. On the contrary, the increase in concentration of D-penetratin resulted in a decrease in insulin absorption into the nasal mucosa. Additionally, there was no histological damage observed in L-penetratin-treated nasal membranes. The authors concluded that L-penetratin could be an effective permeation enhancer for insulin through the nasal mucosa [87]. 

The absorption enhancement of a novel L-penetratin analog, ‘shuffle (R, K fix) 2′, was investigated with insulin, along with other peptides, through the nasal membrane. It was noted that coadministration of insulin with shuffle (R, K fix) 2 resulted in a greater enhancement of insulin across the nasal membrane of rats as compared to L-penetratin alone. This result was presumably due to intermolecular binding between insulin and shuffle (R, K fix), which was higher than the binding between insulin and L-penetratin. Hence, the authors of this study concluded that the binding ratio of a CPP and insulin or any peptide drug is essential for enhancement into the nasal membrane [88]. Following this finding, in silico analysis of penetration analogs predicted a novel CPP (PenetraMax) with a better enhancing effect as compared to shuffle (R, K fix) 2. Therefore, this sequence was investigated. In addition to its ability to deliver insulin through the nasal cavity, it was observed that the bioavailability of nasal insulin at a concentration of 2 mM was 2.3 times higher than L-penetration at the same concentration, which was 100% relative to subcutaneous insulin. Furthermore, it was also determined that PenetraMax, as well as L-penetratin, does not possess any form of toxicity either to the nasal mucosal membrane or the systemic circulation [89]. 

These findings suggest the potential use of the intranasal route, taking into consideration on that the function of enhancers did not, in any way, interrupt the action of insulin.

#### 5.1.2. Buccal

Another alternative is the use of the buccal route for the delivery of insulin. This route has similar advantages to those of the nasal route in terms of the non-invasiveness of the delivery method. Furthermore, the buccal route offers a large, highly vascularized surface area, allowing for permeation of drugs to the systemic circulation system [90]. This results in quick onset of action as compared to other delivery routes. However, disadvantages may include the discomfort of leaving the drugs in the buccal area for a prolonged time. This route of administration may also cause irritation to the buccal cavity, leading to accidental swallowing. Regardless, buccal administration of drugs—especially proteins—is promising due to the minimal invasiveness of this route as compared to injectable agents [91].
In Vitro

Several studies have been carried out to evaluate the delivery of insulin across the buccal membrane. A study was conducted to investigate the use of bile salt sodium glycodeoxycholate (SGDC) as a hydrophobic ion-pairing (HIP) nanocomplex (C1 and C2) to delivery insulin across the buccal membrane. HIP nanocomplex C1 achieved improved permeation of insulin across TR146 cells, as well as porcine buccal tissues, which could be due to the presence of a higher concentration of SGDC at 5.17 mM as compared to HIP nanocomplex C2 with concentration of 1.04 mM [92]. The authors of another study investigating the effect of elastic bilosomes with various derivatives of cholic acid, including sodium taurocholate, sodium glycocholate, sodium cholate (SC), sodium glycodeoxycholate (SGDC), and sodium taurodeoxycholate, on the buccal delivery of insulin concluded that SGDC-modified elastic bilosomes were the most effective in permeating insulin across TR146 buccal cells [93].

Insulin–phospholipid complexes (IPCs) with deformable nanovesicles (DNVs) were developed as a model to address issues of the poor permeability of high-molecular-weight therapeutic drugs for buccal delivery. In vitro analysis on porcine buccal mucosa showed that the deposition and permeability coefficient of insulin in IPC-DNVs were much higher as compared to conventional nanovesicles (IPC-NVs). This is presumed to be due to the presence of deformable vesicles, which allowed them to penetrate across the epithelium. 

Interestingly, another study demonstrated the permeability of insulin with the use of a buccal patch comprising ionic liquid (IL) choline bicarbonate and geranic acid (CAGE) as a permeation enhancer, chitosan as an anchoring agent to the buccal tissue, and insulin and ploy(vinyl alcohol) (CPVA). The release profile of insulin from the CPVA patches was investigated under simulated physiological conditions similar to those of the oral cavity, whereby a Transwell plate consisting of an apical chamber, a permeable membrane, and a basolateral chamber was used with the addition of medium to the chamber, mimicking the oral microenvironment. Upon administration of the CAGE/CPVA insulin patch to the Transwell plate, the polyvinyl alcohol in the patch was completely dissolved, allowing the medium to dilute the CAGE gel and allowing for the release of insulin. In the first 15 min, a total of 27% of insulin was released. The concentration of insulin release was constant for the 6 h study period. Subsequently, ex vivo analysis was carried out using porcine buccal tissue, and the CAGE/CPVA insulin patch exhibited enhanced penetration of insulin into the buccal tissue as compared to the CPVA insulin patch, with which the CAGE gel was absent.

Insulin–phospholipid complexes (IPCs) with deformable nanovesicles (DNVs) were developed as a model to address issues of the poor permeability of high-molecular-weight therapeutic drugs for buccal delivery. It was observed that insulin deposition in the mucosa was much higher in the IPC-DNVs as compared to the conventional nanovesicles (IPC-NVs). This indicates that the presence of a deformable structure plays a crucial role in enhancing the permeation of insulin across the epithelium. Furthermore, the use of IPCs was also important, as they contributed to a deeper permeation into the mucosa compared to nanovesicles that do not contain IPCs (INS-DNVs) [94].

The use of cell-penetrating peptides (CPPs) has been shown to ease the delivery of large molecules into cells, and the use of conjugate may aid in the permeability of insulin across the buccal mucosa. Hence a CPP conjugate (INS-PEG-LMWP) was developed in order to determine the permeability of insulin across the buccal membrane and the relative bioavailability of the complex. The use of INS-PEG-LMWP conjugate enhanced the delivery of insulin across the buccal mucosa as compared to the CPP–insulin mixture and insulin solution. INS-PEG-LMWP resulted in sustained hypoglycemia with a relative bioavailability of 26.86% as compared to the insulin solution [95]. 

A chitosan-based electrospun fiber scaffold was developed using polyethylene oxide (PEO) with varying concentrations to determine the buccal permeability of insulin. It was observed that the content of chitosan plays a significant role in ensuring the quick release of insulin, and a high level of chitosan was observed to induce the fastest release of insulin. A buccal permeation study highlighted that a higher chitosan level enhanced the permeability of the buccal membrane by 16 times relative to that of free insulin. Additionally, this study proved that there was no loss in the bioactivity of insulin throughout the preparation process of the chitosan fiber [96].

Another chitosan-based nanoparticle with insulin (IN-CS-NP) was formulated using the ionic gelation method with sodium tripolyphosphate as a crosslinker. In vitro analysis of IN-CS-NP-loaded buccal film showed that the release of insulin was dependent on the swelling and erosion of polymer [97]. The thiolation of chitosan increases the effectiveness of mucoadhesion and promotes the permeability of chitosan. Furthermore, the enhancing effect of mucoadhesion could be due to the interaction between the thiol group of chitosan particles and the cysteine-rich glycoproteins in the mucus. Hence, in this study, a quaternized ammonium of chitosan, triethyl chitosan (TEC) with the addition of L-cysteine (Cys), was synthesized to form nanoparticles. In a simulated buccal environment with a pH of 7.4, the release of insulin was rapid for the first 30 min but was slowly sustained in a controlled manner up to 480 min, with an average of 98% of insulin released. Ex vivo analysis of TEC-Cys on rabbit buccal tissue showed a burst of insulin release for the first hour of the experiment, reaching a plateau within 2 to 4 h. Subsequently, an increase of up to 96% was observed within 4–8 h. The authors concluded that the thiolation of TEC-Cys was successful in enhancing the permeation of insulin across the buccal mucosa [98].

Similarly, in another study, insulin-loaded mucoadhesive buccal films (MBFs) with the addition of glycerin and L-arginine exhibited good control of blood glucose reduction for 8 h in diabetic rats [99]. Additionally, research was conducted on the permeability of insulin across buccal cells, with the enhancement of amino acid. It was hypothesized that amino acids are less toxic as compared to sodium deoxycholate, a bile salt that is commonly used as a permeation enhancer. According to the results, amino acids (lysine, histidine, glutamic acid, and aspartic acid) showed a good permeation of insulin across TR146 buccal cells [100].
In Vivo

Following the successful delivery of insulin using an SGDC-modified elastic bilosome, a study was conducted to investigate the effect of SC-incorporated elastic liposomes (SC-EL) and SGDC-incorporated elastic liposomes (SGDC-EL) on the permeation of insulin across porcine buccal tissues. The use of SGDC-EL enhanced the delivery of insulin across the buccal membrane as compared to SC-EL. This was hypothesized to be due to the lipophilic and enhancing properties of SGDC. It was also noted that a higher level of deformability results in improved internalization of vesicles across the membrane and that the deformability level of SGDC-EL was higher than that of SC-EL [101]. 

A CAGE/CPVA insulin patch was further tested on an in vivo model, and sustained hypoglycemia was observed in rats with varying doses of insulin. A similar pattern of blood glucose level was observed in the first three hours for both the 7.4 and 15 U kg^−1^ doses, with a moderate drop in the percentage of blood glucose. However, there was a further reduction of up to 50% of blood glucose by the end of 6 h with a dose of 15 U kg^−1^. Additionally, no significant damage to the organ and tissues was noted in rats following the application of CAGE/CPVA patches. This study proved that a polymeric l-based patch made of CPVA/CAGE gel may be a suitable candidate for the delivery of insulin across the buccal membrane [102]. 

The use of deformable vesicles for delivery of insulin considerably enhanced the permeability of insulin across the buccal membrane. Therefore, an in vivo analysis was carried out in normal rabbit to determine the relative pharmacological activity of insulin delivered through PIC-DNVs, INS-DNVs, and IPC-NVs, indicating that IPC-DNVs had sustained hypoglycemia as compared to INS-DNVs and IPC-NVs, with a high relative bioavailability (15.53%) as compared to that of NS-DNVs (3.09%) and IPC-NVs (1.96%). 

Following the in vitro release study of insulin from IN-CS-NP–buccal films, in vivo analysis indicated a decrease in the level of blood glucose to 52.2% of the reference film administered within 5 h. The authors of this study confirmed the efficacy of the use of buccal film loaded with INS-CH-NPs, resulting in the controlled release of insulin, as well as a significant reduction in the level of blood glucose in rats [97]. 

The evidence presented above shows that insulin delivery is possible through buccal tissues; however, further studies are warranted to investigate the long-term safety of the proposed complexes. It is also important to elucidate the mechanisms of action of the complexes, as they could contribute to the understanding of other potential complexes and aid in the development of superior carriers for permeation into the buccal epithelium. However, in all the mentioned studies, no observable irritation or alteration was reported.

#### 5.1.3. Oral

Insulin delivery via the oral route has been found to be advantageous due its non-invasive administration; however, as a result of the harsh stomach environment, the use of pure insulin for oral delivery is challenging due to the possibility of degradation across the gastrointestinal tract. Additionally, due to its large molecular size, insulin may not be readily absorbed across the oral membrane [103]. These limitations have been discussed in many studies in order to achieve efficient insulin delivery.
In Vitro

Chitosan-based nanoparticles (Ch-NPs) have been developed with the addition of poly(sodium 4-styrenesulfonate) (PSS) as a crosslinking agent and polyglutamic acid (PGA) for improved oral uptake. An in vitro release study was carried out in three different microenvironments: simulated gastric fluid (SGF), simulated intestinal fluid (SIF), and PBS. Ch-PGA-NPs induced a burst release of insulin in the SGF, probably due to the instability in the gastric environment. However, only a minimal amount of insulin release was observed from the Ch-PSS NPs and Ch-PSS-PGA NPs. In the SIF and PBS, on the other hand, Ch-PSS NPs and Ch-PSS-PGA NPs exhibited a controlled release of insulin. Additionally, Ch-PGA NPs was observed exhibit 50% insulin release in 4 h, whereas the same amount of insulin release was observed with Ch-PSS NPs and Ch-PSS-PGA NPs in 8 h. Subsequently, the intestinal uptake of the developed NPs investigated in Caco-2 cells. The absorption of Ch-PSS-PGA into the intestinal mucosa was significantly higher than that of the Ch-PSS NPs and Ch-PGA-NPs [104].

Molecular factors influencing the transepithelial permeation across the Caco-2 cell monolayer was studied using a common carrier peptide, penetratin, along with its analogs, PenShuf, PenArg, and PenLys. The administration of insulin, along with penetratin, PenShuf, and PenArg, at a pH of 5 resulted in a improved permeation of insulin as compared to a pH level of 7.4 or 6.5. The PenLys analog, on the other hand, did not facilitate the permeation of insulin at any of the tested pH levels. Additionally, increasing the ratio of carrier peptide insulin from the initial 4:1 to 6:1 at a pH level of 5 using PenShuf or PenArg significantly increased the permeability of insulin as compared to the use of penetratin or PenLys. It was presumed that the specific positioning of the Trp residues may have given rise to the increase in interaction on the surface of the Caco-2 epithelium, which promotes permeation. Subsequent evaluation of the cellular toxicity of penetratin and its analog showed no observable decrease in cellular viability, except PenShuf. Cytotoxicity was observed with the ratio used at pH 5, with significantly higher cytotoxicity at a ratio of 6:1, which might infer to an increase in interaction between PenShuf and the surface of the membrane [105].

In another study, Simon et al. demonstrated a coencapsulation of insulin with soybean trypsin inhibitor (SBTI) and sodium caprate (C10) using an inverse-flash precipitation (iFNP) platform, resulting in an encapsulation efficacy of more than 98%. The use of SBTI as a protease inhibitor and C10 as a permeation enhancer could potentially prevent the degradation of insulin within the gastrointestinal tract [106]. Ahmed et al. formulated an encapsulation of insulin (INS) in between layered duple hydroxide (LDH) and chitosan nanoparticles (CSNPs). The entrapment of insulin proved to be effective when there were no observable degradation or reduction in burst release of insulin compared to free insulin. Additionally, the hypoglycemic effect of LDH-INS-CSNPs was significant—comparable to that of oral insulin and subcutaneous insulin [107].

In another study, insulin-loaded, chitosan-based, multifunctional nanocarriers modified by L-valine (LV) and phenylboronic acid (PBA) were evaluated. In vitro analysis showed that insulin release from the CMCS-PBA-LV at a pH of 6.8 is significantly higher at 50.7% as compared to pH 1.2. This could be due to the ability of the carboxyl groups be protonated at a lower pH, which prevents the release of the drug. This indicates that nanocarriers can be fabricated in response to the change in pH from the stomach to the intestine without any form of insulin degradation during the delivery process [108]. 

The development of liquid crystalline nanoparticles (LCNPs) was investigated with respect to the efficacy of insulin upon oral administration. The stability of insulin was observed under simulated biological conditions. An in vitro study showed that insulin was released in a controlled manner over a period of 24 h, indicating the slow dissolvability of LCNPs to release insulin. Additionally, the cellular uptake of insulin was investigated in Caco-2 cells, and it was observed that insulin-loaded LCNPs exhibited a significantly higher uptake into the cells as compared to free insulin. This was in line with the obtained cumulative hypoglycemia, which is higher than that of subcutaneous insulin, highlighting the possibility of LCNPs to exhibit sustained release of entrapped insulin [109].
In Vivo

The positive outcome of chitosan NPs allowed for the subsequent confirmation of their therapeutic efficacy in diabetic animals. Ch-PSS-PGA NPs resulted in 1.7-fold higher cumulative hypoglycemia when compared to subcutaneously administered insulin. This is presumed to be due to the enhanced protection and sustained release of insulin from the NPs, mimicking those of the physiological pattern. This study has also highlighted that the entrapped insulin retained it functionality and did not undergo any sort of degradation in simulated intestinal pH, owning to the presence of PSS and PGA in the Ch NPs [104]. 

Despite its superiority in promoting the penetration of insulin across the transepithelial membrane, PenShuf was not included for the in vivo analysis after exhibiting significant toxicity towards Caco-2 cells. A preliminary study was carried out to evaluate the ability of penetratin to promote delivery of insulin across the intestinal epithelium. The effect of penetratin–insulin complexes at pH 5 was a significant decrease in blood glucose to 50% of the initial level between 30 and 60 min. However, at pH 7.4, there was no difference in reduction in blood glucose when compared to administration of insulin alone.

A hypoglycemic study was carried out to further evaluate the pharmacological effect of CMCD-PBA-LV. It was observed that the oral delivery of insulin-loaded nanocarriers resulted in blood glucose depression of 60.2% after 5 h, with a slow increase to its initial baseline at 8.4 h, which was significantly higher than that of subcutaneous insulin, for which the reduction in glucose level was observed within the first 2 h, with an increase in blood glucose within 3.4 h of administration. The study authors concluded that L-valine acts as a mucoadhesive, adhering to the gastrointestinal membrane, allowing for slow release of insulin [108].

CaCO3-based composite nanocarriers (NCs) with hyaluronic acid (HA) coatings were employed due to their biocompatibility and pH sensitivity. The oral administration of CaCO3-HA NCs resulted in hypoglycemia in a controlled-release manner as compared to subcutaneously administered insulin. Additionally, the formulated nanocarriers were able to internalized into the HT-29 cells lines for the release of insulin within the cells for more than 6 h [110].

Insulin-loaded poly(lactic-co-glycolic) acid (PLGA) and folic-acid-modified chitosan (FA-CS) nanocarriers was fabricated via a self-assembly method. The formulated nanocarriers exhibited enhanced entrapment efficacy of insulin under simulated biological conditions as compared to free insulin. An in vivo study showed that insulin-loaded PLGA-FA-CS nanocarriers were able to induce hypoglycemia in a sustained manner as compared to subcutaneously administered insulin, following which a spike in hypoglycemia was observed within two hours after administration. This study suggests the potential use of composite nanocarriers as a drug carrier for the oral delivery of insulin [111]. Despite many efforts, it is challenging to find an effective and safe method for the oral delivery of insulin. However, with continuous research, there is a possibility of attaining a more successful formulation for delivery of not only insulin but many other peptides in a non-invasive manner. 

#### 5.1.4. Transdermal

Transdermal delivery is alternative method that can be employed to non-invasively deliver insulin across the skin membrane. Transdermal patches have been used for almost 40 years. In that sense, it is safe to assume that transdermal delivery may potentially promote the delivery of insulin in a painless manner. They can also avoid first-pass metabolism, which could enhance the bioavailability of insulin. However, the downside of this delivery method is the need to formulate an encapsulation or enhancement that aids in the penetration of insulin across the skin membrane. Various studies have been employed to overcome these limitations.
In Vitro

Ligheswar et al. studied the efficacy of various concentrations of polymers in enhancing the permeability of chitosan–insulin nanoparticles. This study revealed that patches formulated with polymers such as polyethylene glycol (PEG) or hydroxypropyl methylcellulose (HPMC) are able to enhance the penetration of insulin across the skin. Additionally, patches with a concentration of 60% HPMG and 40% PEG were proven to promote the best insulin permeability as compared to other combinations of concentrations of both the polymers. The thermodynamic activity of patches was assumed to be due to the presence of functioning glycols, along with the combination of solvents in the formulation [112]. 

In another study, the enhancement activity of insulin in a solid-in-oil (S/O) nanodispersion was investigated using protein transduction domains (PTDs). Owing to their ability to induce permeation for transcutaneous protein delivery, PTDs have been investigated with the addition of various recombinant proteins. Oligo-arginine peptides were used as PTDs, along with isopropyl myristate (IPM). It was observed that S/O nanodispersion with the addition of oligo-arginine enhanced the permeation of insulin into the skin. The concentration of insulin across the skin was higher with the use of S/O nanodispersion with R6 peptides. Moreover, it was highlighted that the involvement of IPM and PTDs aided in the alteration of the skin barrier, which enhanced the transdermal ability to deliver insulin using the S/O nanodispersion with arginine-rich peptides [113,114]. 

Chang et al. determined the permeation-enhancing activity of cationic cyclopeptides based on a sequence of TD-1 partially substituted with arginine or lysine on Caco-2 cell monolayers. It was noted that amongst the tested cyclopeptides, TD-34 with substituted lysine at the N-5 and N-6 positions exhibited the best enhancement activity. It was also observed that TD-1 peptide may improve the transdermal absorption of insulin by disrupting the follicle of the epithelial tissues. Additionally, it should be noted that the use of Caco-2 cell monolayers (BL→AP) could be adopted as a potential preliminary method to determine the transdermal activity of peptides with respect to the absorption of insulin due to its comparable similiters to percutaneous insulin absorption [115]. 

A total of 43 functional groups were identified for their enhancement properties with respect to insulin permeation were. In this study, a virtual design algorithm, in combination with a quantitative structure–property relationship (QSPR), was used to predict the properties of these chemical permeation enhancers (CPEs). CPEs were screened using the change in electrical resistance, an alternative method to the laborious traditional method; of the tested CPEs, 22 were chosen for further analysis. This study revealed that no specific functioning groups are responsible for the enhancement of insulin permeation; instead, CPEs with a low positive log Kow and at least one hydrogen donor or acceptor (toluene being an exception) are considered to provide good permeation enhancement. However, the factor that limits the use of these CPEs as an enhancer is toxicity, as only eight of the 22 tested CPEs were found to be non-toxic [116].

In another study, Tanner et al. established the ratio of an ionic liquid, choline bicarbonate, and geranic acid (CAGE) necessary for the transdermal delivery of insulin. In this study, the ratios used for the synthesis of CAGE were 1:2,1:4, 2:1, and 1:1. It was noted that the variant of CAGE with ratios of 2:1 and 1:1 showed detectable fluorescence in the epidermic and dermic with no effect in transport of insulin across the SC, which was similar to skin treated with PBS as a control. The CAGE-variant ratios of 1:2 and 1:4 resulted in a remarkable fluorescence intensity across the skin. This was hypothesized to be due to the presence of excess geranic acid during the synthesis of CAGE variants. However, neither the skin treated with only choline bicarbonate nor that treated with geranic acid exhibited any transport of insulin across the SC. The authors concluded that the synthesis of IL is crucial in order to aid in the facilitation of insulin across the skin [117].
In Vitro

Similarly, the use of gold nanorods (GNRs) that formed a complex with an edible surfactant and insulin (INS) in an oil phase to form a solid-in-oil (SO) formulation (SO–INS–GNR) were studied. The ability of insulin to penetrate well through the skin with SO-INS-GNR with the addition of near-infrared light (NIR) light irradiation as compared to the absence of NIR light irradiation was investigated. Results showed a strong fluorescence intensity of FITC-labelled insulin in rats with the addition of NIR light irradiation compared to when NIR light irradiation was absent. This implies that after irradiation, gold nanorods in the complex absorbed NIR light and then converted light energy into heat that could break the SC of the skin [118].

Transdermal insulin delivery serves as a potential alternative to existing subcutaneous delivery methods due to its vast use of various enhancers and techniques [119,120,121,122,123], which has successfully delivered insulin across the skin membrane, taking into consideration that bioavailability and functionality of insulin were not altered. 

#### 5.1.5. Vaginal

Vaginal insulin delivery is another potential route for drug delivery. Vaginal tissue has a large surface area with abundance of blood vessels that could aid in the systemic delivery of treatments.
In Vitro

Niosomes have been adopted as drug carriers to enhance permeation, owing to their various benefits [124] ( A. Similarly, in this study, sorbitan monoester (insuli-Span 40 and -Span 60 niosomes) was used to form niosomes to deliver insulin vaginally. in vitro release study of niosomes–insulin Span 40 vesicles showed a quicker release of insulin in simulated vaginal fluid (acetate buffer at pH 4.5) as compared to Span 60 vesicles. After 24 h of study, it was observed that roughly 30% of the originally entrapped insulin was released, suggesting that that noisome vesicle may be used to deliver insulin in a long-term and controlled manner.

Chitosan ascorbate nanoparticles (NPs) encapsulated in hydrophilic freeze-dried cylinders were formulated to deliver insulin intravaginally. In this study, a hydrophilic freeze-dry system was chosen for its ability to ensure the quick release of various peptide-loaded nanoparticles into the vaginal area. A combination of mannitol (Ma)/sucrose (Sa) and gelatin (GeB) was found to be the most effective hydrophilic agent employed in this study due to its considerable ability to resist mechanical rupture while successfully releasing NPs without altering their size and morphology. An in vitro release study indicated that 50% of insulin was released in the first 15 min, with almost 100% released after 30 min, indicating that the chitosan ascorbate NPs were able to release insulin into the vaginal mucosa. This was further supported by ex vivo analysis, which revealed that insulin into was able to penetrate the porcine vaginal membrane at a depth of 50–55 μm as compared to 20–25 μm for insulin solution alone. Therefore, the encapsulated insulin–chitosan ascorbate NPs (Ma/Su/GeB) were investigated to determine the capability of a freeze-dried cylinder to release insulin-loaded NPs into the vaginal mucosa. It was observed that insulin penetrated across the vaginal membrane to a depth of 50 μm, which is similar to that of insulin-loaded NPs. The authors concluded that a freeze-drying system consisting of Ma/Su/GeB is a suitable option for the delivery of peptides into the vaginal mucosa [125].
In Vivo

Insulin–chitosan gel with the addition to two other enhancing agents, taurocholate (TAU) and dimethyl-β-cyclodextrin (DM-βCD), was administered to the vagina and rectum of rats. The hypoglycemic level was observed to be higher in rats that were given DM-βCD-chitosan gel both through the vaginal and rectal routes. It was also observed that DM-βCD-chitosan gel resulted in a larger opening in the intercellular space as compared to TAU-chitosan gel or chitosan gel only. This shows that DM-βCD is a potential enhancing agent for the delivery of insulin across through vaginal and rectal routes [126]. 

Based on the success of controlled release of insulin from Span 40 and Span 60, an animal study was carried out in diabetic rats. After 1.5 h, maximal hypoglycemia was observed following vaginal administration of insulin-Span 40 and -Span 60. Even after 6 h, the level of plasma glucose was lower than the initial level of blood glucose, further proving that noisome insulin vesicles may promote a sustained and long-term hypoglycemic effect [127]. 

#### 5.1.6. Rectum

The delivery of insulin using the rectal route could also be effective due to high vascularization, which allows drugs to be absorbed and transported to the systemic circulation system. This provides a significant advantage, as drugs could avoid the first-pass effect. This route is also beneficial in that the administered drug is not be degraded by the harsh environment and enzymes produced by the body. However, the disadvantage of this route is the presence of a thick coating along the mucosal membrane protecting the epithelial wall, creating a membrane barrier [128]. This issue can be overcome by developing an enhancer that would aid in the permeability of insulin across the protective barrier.
In Vitro

A dimple suppository was developed using Labrasol as an absorption enhancer in order to deliver insulin across rectal tissues. This study successfully proved that Labrasol aids insulin delivery across rectal tissues. It was also highlighted that limiting the surface area of insulin in the suppository helped to enhance rectal delivery because the surface area of a one-dimple suppository compared to a three-dimple suppository was smaller and covered in hard fats, except at the site of sealing. This allowed the insulin to only be delivered to the rectal mucosa without the interference of other fluid in the mucosal site [129]. 

One of the advantages of rectal delivery is that the pH of the rectum is similar to that of the internal environment, which prevents the alteration of drugs upon administration. A novel hydroxypropyl methyl cellulose-co-polyacrylamide-co-methacrylic acid (HPMC-co-PAM-co-PMAA) hydrogel was developed with insulin (INS) to be used as a rectal suppository. An in vitro release study was performed in a simulated microenvironment with a pH of 7.4, and after 24 h, 72.6% of encapsulated insulin was detected in the solution, indicating that INS-HPMC-co-PAM-co-PMAA could prompt the release of insulin in the rectum, as the normal pH level of the rectum is known to be 7.4 [130].
In Vivo

The efficiency of insulin release from an INS-loaded HPMC-co-PAM- co-PMAA hydrogel in a simulated rectal microenvironment prompted a subsequent hypoglycemia study. INS-loaded HPMC-co-PAM-co-PMAA resulted in a similar drop in blood glucose to that of insulin injection within the first 1 h. After 7 h of treatment, the glucose level of rats given insulin injection recovered slightly, but the same was not observed in rats with INS-loaded hydrogels. This indicates that the INS-loaded hydrogel may promote the sustained release of insulin via a rectal suppository [130].

A novel polyacrylate methacrylic acid-co-hydroxyethyl methacrylate-co-methyl acrylate (MAA-co-HEMA-co-MA) copolymer hydrogel was formulated and dissolved into methylcellulose (MC) for the delivery of insulin (INS) as a rectal suppository. The hypoglycemic level of diabetic rats treated with INS-loaded binary hydrogel suggested a significant decrease in blood glucose at 7.8 mmol/L as compared to subcutaneous insulin, which resulted in a 10 mmol/L decrease within 4.5 h. A continuous drop in glucose level was observed in rats treated with INS-loaded binary hydrogel, even after 8.5 h [131].

### 5.2. Physical Enhancers

#### 5.2.1. Iontophoresis

This non-invasive method of drug delivery brings about many advantages over the existing conventional routes. Iontophoresis is a less painful method of administration over a long period of time and could avoid first-pass metabolism. There is a low chance of infection and overdosing, as well as higher chances of compliance amongst patients. However, these modes of delivery have similar shortcomings in terms of the ability of drugs to be absorbed across the membrane barrier, reducing the dose of drugs that can be delivered at each point in time. Although the end motif of iontophoresis is similar to that of chemical enhancers, the outcomes differ in terms of the insulin delivery method. Iontophoresis uses electrical force to temporarily disrupt the stratum corneum of the skin for the delivery of insulin [132]. A study was conducted to investigate the potential enhancement of iontophoresis in delivery of insulin nanoparticles across intestinal cells. An in vitro permeation study indicated that iontophoresis aided in the permeation of insulin across the intestinal membrane as compared to passive diffusion. An in vivo permeation study further justified the effectiveness of iontophoresis in consistently delivering insulin across the intestinal membrane [133]. Similarly, the use of iontophoresis across intestinal cells resulted in improved paracellular transport of insulin due to the opening of the tight junctions. A subsequent in vivo study proved that iontophoresis induces hypoglycemia in rats treated with insulin-loaded mucoadhesive patches. There was no observable damage to the intestinal wall after the administration of iontophoresis, indicating the safe use of this technology [134].

#### 5.2.2. Microneedling

Microneedling is a minimally invasive method that can be applied for the delivery of insulin. Microneedles are designed in such a way that allows them to penetrate through the stratum corneum of the skin for a rapid release of drugs without causing permanent damage to the skin. Currently in the pre-clinical study phase, microneedles are classified into a few categorized based on the type and materials they are made of. As such, solid microneedles have been developed to create an opening that is sufficient to deliver small molecules and proteins. Studies have proven that solid microneedles are capable of penetrating human skin. They are also capable of delivering insulin across the skin, reducing the level of glucose in diabetic rats [135,136]. Li et al. evaluated different dimensions of polymer microneedles (MNs) for the enhancement of transdermal drug delivery. The best mechanical stability was achieved with microneedles with a length of 600 μm as compared to 700 and 800 μm. The percentage of penetration of MNs with a length of 600 μm remained at 90%, even after repeated penetration into the skin (20th insertion). As for the 700 and 800 μm microneedles, the amount of successful insertion reduced sharply to less than 20%. This was assumed to be due to the possibility that MNs were easily bent with a longer body as compared to a shorter body, leading to improved mechanical stability. However, the dimensions of MNs play a larger role by allowing for more drugs to be permeated into the skin. Skin pre-treated with MNs resulted in higher permeation of drugs, especially with MNs with a length of 700 and 800 μm, as compared to no pre-treatment. The authors concluded that longer MNs can induce a deeper and larger microhole, which ultimately leads to enhanced penetration of drugs into the skin. An in vivo absorption study of rats pre-treated with MNs showed a reduction in blood glucose level to 29% from its initial 100% after 5 h as compared to subcutaneous injection, which resulted in a rapid reduction in blood glucose level from 100% to 19% within 1.5 h. Subcutaneous release of insulin occurs much more quickly than that of MNs, which release insulin in a controlled manner [137].

Another alternative is the use of dissolving microneedles, whereby drugs are encapsulated within a soluble matrix upon insertion into the skin. Liu et al. evaluated the effect of the use of hyaluronic acid in the preparation of insulin-loaded microneedles. Microneedles completely dissolved 1 h after administration. Transepidermal water loss was observed upon administration, although water levels returned to normal after 24 h, indicating that the alteration of the skin was reversible. Another study was focused on developing a microneedle patch made of gelatin and starch, resulting in good dissolution, with needles completely dissolved within 5 min of administration [138]. An ability to reduce plasma glucose was observed in both of these studies. 

Chen et al. suggested the use of poly-γ-glutamic acid (γ-PGA) MNs and polyvinyl alcohol (PVA)/polyvinyl pyrrolidone (PVP) as a supporting structure for the development of fully insertable microneedles. Upon administration, the microneedles were dissolved within 4 min, allowing for the release the entire drug load, suggesting that the use of a supporting structure for the development of microneedles may aid in a quick release of drugs without the alteration of pharmacological activity. An in vivo hypoglycemic study showed that microneedles were able to reduce plasma glucose levels similarly to subcutaneously administrated insulin [139]. Lau et al. also investigated the use of multi-layered dissolving microneedles composed of silk fibroin supported on a flexible polyvinyl alcohol pedestal for the delivery of insulin. The fabricated needles were able to be inserted and induced hypoglycemia through the abdomen skin of mice [140]. Two-layer dissolving microneedles composed of sodium carboxymethyl cellulose (CMC) and gelatin were investigated. The gelatin/CMC microneedle patches enhanced the delivery of insulin with a relative bioavailability of 85.7% [141]. Another formulation of two-layer dissolving polyvinylpyrrolidone (PVP)-based microneedles with varying molecular weights was developed for the delivery of insulin in vivo. Of the tested molecular weights, PVP, PVP10, and PVP360 with a ratio of 1:3 were found to successfully enhance the in vivo delivery of insulin [142]. 

Another alternative to dissolving microneedles is biodegradable microneedles, for which the release of drugs from the microneedles matrices is controlled and can be sustained for a long period of time. They are also inexpensive and safer compared to the other types of microneedles.

An example of biodegradable microneedles is a study carried out by Yu et al. on biodegradable microneedles consisting of gelatin/calcium sulfate hemihydrate (GelCS) composites for the delivery of insulin. The fabrication process did not cause denaturation to insulin encapsulated into the microneedle; however, the biological activity of insulin was reduced when it was dissolved at a higher temperature of 45 °C as compared to 30 °C. However, this study demonstrated that the fabrication process of the microneedle itself is safe for the encapsulation of insulin. Additionally, GelCS microneedle patches with insulin were evaluated in diabetic rats. Subcutaneously administered insulin reduced the level of blood glucose within 1 h (47 md/dL), followed by an increase to the initial level after 6 h. However, rats treated with insulin-loaded microneedles showed a slow drop in blood glucose levels (90.2 mg/dL in two hours), followed by a slow increase to the initial glucose level. The amount of insulin was increase from 5 IU to 10 and 20 IU to be loaded into the microneedle, which resulted in a rapid decrease in blood glucose level, followed by a slow increase to the initial level. The relative bioavailability of insulin was further analyzed. Following subcutaneous administration, the plasma insulin level reached a peak value of 130 μIU/mL in 1 h, returning to its initial state within 4 h. On the other hand, insulin-encapsulated biodegradable microneedles resulted in the highest serum insulin level (230 μIU/mL) using 20 IU insulin in 1 h as compared to 5 IU. This is because insulin permeating across the skin barrier from the microneedle may require a longer period of time to reach the systemic circulation system. Hence, the use of a higher dose of insulin in the microneedle would allow the serum insulin to be maintained at a higher level for a longer period of time. The authors of this study concluded that biodegradable microneedling is a potential application for the treatment of diabetes mellitus with a better outcome in maintaining serum insulin for a longer period of time as compared to subcutaneous insulin administration [143].The use of microneedles is a suitable substitute for therapeutic delivery of insulin. Considerable efforts have been made to discover different types of microneedles to successfully deliver insulin across the skin [144,145,146,147]. However, attention should be given to the possibility of irritation, swelling, and erythema, as well as the retention of microneedle particles, which could represent a serious obstacle to the effective delivery of insulin [148]. 

#### 5.2.3. Inhalation

The pulmonary route of several advantages over other therapeutic routes. The large surface area of pulmonary tissue provides an opportunity for drug permeation. A study on the use of aerosolized insulin was conducted in the 1920s, and surprisingly, the authors were able to reduce the level of blood glucose. However, due to the low bioavailability of insulin, this method was not successful [149]. A newer inhalation system was developed with better technology and true enough, with the first inhaled insulin therapy, Exubera, reaching the market in 2006. An in vitro study showed that Exubera was able to deliver at least 50% of the loaded insulin dose, depending on the fill weight; a higher fill weight can release a larger dose as compared to a lower fill weight [150]. Despite this, Exubera failed when it was withdrawn shortly after its introduction due to economical constraints that negatively impacted profit margins [151]. Various presumptions were made with regard to the withdrawal of Exubera, including concerns surrounding the bioavailability of insulin after each administration, which was less than that of subcutaneous insulin, requiring multiple does of insulin to achieve optimal insulin action [152]. Another concern was that 6 out of 4740 patients who used Exubera in the clinical trial phase developed lung cancer; however, all of the patients diagnosed had a previous history of cigarette smoking, and the number of cases reported is too low to be able to report on the association of Exubera with the development of lung cancer [153]. Despite the small number of incidences, there were still doubts in terms of safety, as well as acceptance by patients and physicians. 

## 6. Advancement of Alternative Routes for Insulin Delivery to Clinical Trials

Considerable efforts have been made with respect to the discovery of a non-invasive method of insulin delivery (Table 3). Among the wide range of studies conducted to this end, the automated insulin delivery (AID) (Figure 3) system stands out as one of the most interesting techniques employed for the delivery of insulin. Not only is an AID able to deliver insulin, but it can monitor the level of glucose in order to control the delivery using a dosing algorithm [154]. In that light, several AID systems are currently in various stages of clinical trials. An early-phase feasibility study using a Lilly hybrid closed-loop (HCL) system composed of an investigational insulin pump, insulin lispro, a pump-embedded model-predictive control algorithm, a continuous glucose monitor (CGM), and an external dedicated control results in a safe and satisfactory glycemic control in response to stimulated diabetes management challenges (NCT03849612, NCT03743285) [155]. The treatment guidelines for hypoglycemia recommends the use of 15–20 g of carbohydrates; however, the guidelines do not account for a reduction in insulin suspension by predictive low-glucose suspend (PLGS). This study showed that rescue treatment after insulin suspension in a situation of hypoglycemia only consisted of 9 g of carbohydrates with no repetition of carbohydrate treatments. Hence, the amount of carbohydrates needed for the treatment of hypoglycemia using the PLGS system should be revised (NCT03890003) [156]. Intradermal insulin delivery techniques using a hollow microneedle have been studied in children and adolescents with type 1 diabetes mellitus. As hypothesized, microneedle-based insulin delivery was able to deliver insulin with less insertion pain and faster onset and offset in children and adolescents. This could aid improve compliance with insulin delivery (NCT00837512) [157]. A clinical trial on an insulin infusion device, PaQ, was carried out, and the feasibility of its use was confirmed in patients with type 2 diabetes who had been using multiple daily injections (NCT01535612) [158]. A similar study of the PaQ (a wearable device) was carried out to determine basal rates and bolus insulin on demand; adults with type 2 diabetes showed improved glycemic control using a PaQ device and were satisfied with the change from daily injections (NCT02419859) [159]. iLet is a new type of insulin system with a CGM that is able to detect the glycemic level in the body and deliver a controlled amount of insulin needed at a given time. However, in this study, a fast-acting insulin aspart was used to determine the safety profile in the original setting of the iLet system, and no safety concerns were observed with this change (NCT03816761) [160]. Another artificial pancreas system that could achieve constant glucose monitoring for controlled insulin delivery upon detected patterns of change in glucose has been studied (NCT01484457). The newly FDA approved closed-loop insulin delivery system has been employed to track on the initiation of the first year of clinical use. Although these systems have been associated with good glycemic control, several patients who had previously received the 670G system have discontinued its use due to concerns over sensor issues, hypoglycemia, problem with obtaining sources, preference for multiple injection, and sports. There were also significant concerns related to the use of auto mode to control the level of hemoglobin A1c during follow-up visits (NCT03017482) [161]. A closed-loop therapy was also studied in children younger than 7 years old with type 1 diabetes. The closed-loop insulin delivery technique decreased the severity of overnight hyperglycemic incidents without inducing hypoglycemia. The glucose levels in children almost reached a target before their next meal. This implies the benefit of using such a system for children with type 1 diabetes (NCT01421225) [162]. Another bio-inspired artificial pancreas (BiAP) successfully achieved safe glycemic control during fasting, as well as overnight and postprandial conditions (NCT01534013). Twenty subjects were selected for a 6 h fasting closed-loop study. The primary outcome of the fasting study was that 98% of the time was spent in the target range (3.9–10.0 mmol/L) and the euglycemia range (3.9–7.8 mmol/L), which was comparable to results obtained in other closed-loop fasting studies. Additionally, none of the time was spent in a severe hypoglycemia or hyperglycemia state. A sub-analysis of the first 2 h of study indicated that 89% of subjects were within the target range, which indicates the ability of the control to reduce the level of glucose to the target range, regardless of the initial condition. Hypoglycemia was observed in one subject, but no episodes of severe hypoglycemia were noted. Overall, this study confirmed that the BiAP system is safe for use [163].

## 7. Conclusions

In this review, we substantially discussed the possible alternative routes for insulin delivery that are non-invasive and patient-friendly with the intention of replacing existing subcutaneous insulin injections. These methods include chemical and physical enhancers, both of which provide improved diabetes care through improvements in efficacy and safe delivery of insulin into the systemic circulation. Chemical enhancers of insulin delivery via the nasal, buccal, oral, transdermal, vaginal, and rectal routes have achieved promising results with respect to the delivery of insulin. Intranasal and buccal delivery methods are considerably effective and well-studied, but given the route of administration, there could be potential pain and discomfort at the target site if used for prolonged periods. Ideally, oral uptake of insulin should be considered as an alternative route to subcutaneous insulin injections. However, poor insulin bioavailability and is tendency toward enzyme degradation are the main concerns; therefore, more clinical studies are required to overcome these challenges. Transdermal, vaginal, and rectal delivery of insulin have also been explored, causing less discomfort for patients, although such delivery methods require external enhancement to penetrate the epithelial membrane. Physical enhancers of insulin delivery are a new and promising approach for non-invasive administration. However, more studies are required to understand insulin stability and bioavailability when administered with such methods. 

At present, it is not possible to conclude with certainty that the non-invasive methods discussed in this review are preferable to traditional insulin delivery, as none of the novel technologies have superseded the trusted subcutaneous administration. Nevertheless, such non-invasive delivery methods are attractive due to their ability to avoid pain and their ease of administration. Each route and delivery method have their respective advantages and disadvantages, and it is not yet possible to achieve universal acceptance of such delivery systems. However, non-invasive approaches to insulin delivery will redefine current treatment of diabetes and will be widely acceptable to patients and healthcare institutes, particularly if they are cost-effective and patient-friendly.

## Figures and Tables

**Figure 1 pharmaceutics-14-01406-f001:**
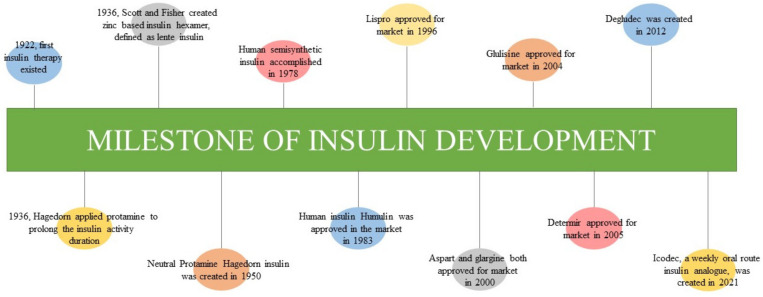
Timeline of the evolution of diabetes management from 1922 to 2021.

**Figure 2 pharmaceutics-14-01406-f002:**
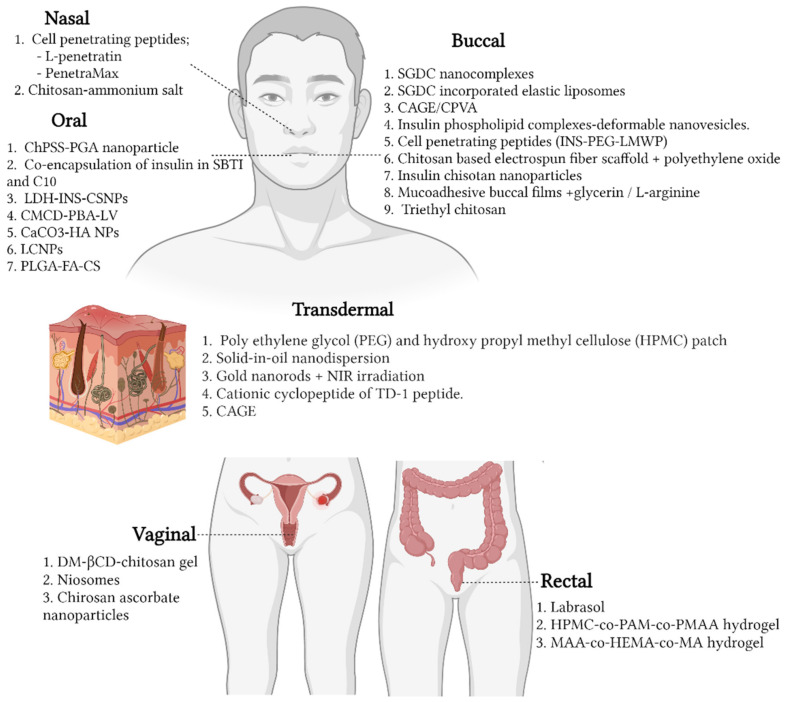
Potential alternative routes for the delivery of insulin, such as oral, nasal, buccal, transdermal, vaginal, and rectal routes, as well as the types of chemical enhancers that have been used in pre-clinical settings for the successful delivery of insulin via the chosen routes.

**Figure 3 pharmaceutics-14-01406-f003:**
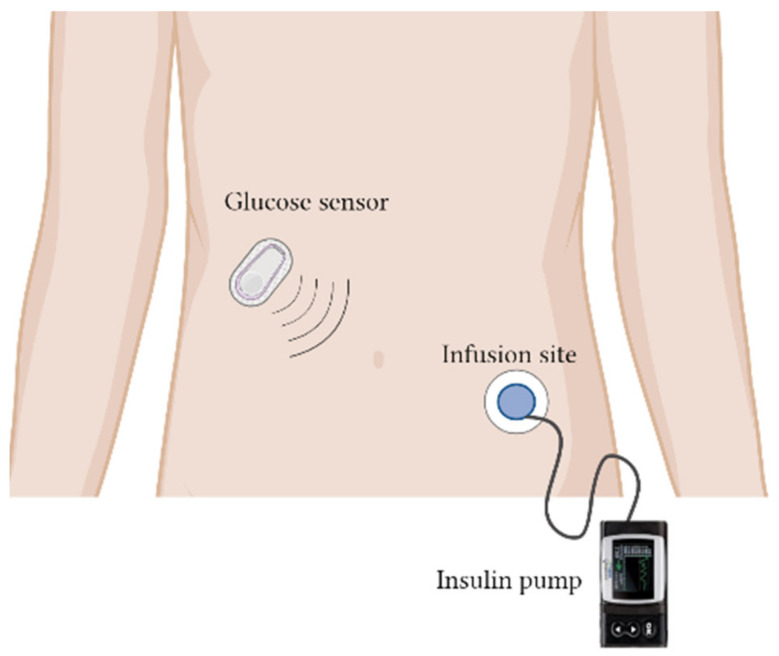
**Automated insulin delivery system conducted in a** closed-loop in between glucose sensing device and insulin delivery device (Infusion site) for delivery of insulin. Briefly, the components of a closed-loop insulin system consist of a glucose sensor that measures the interstitial glucose level, which is then transmitted to the control algorithm (on a smartphone or an insulin pump). The control algorithm is able to compute the amount of insulin to be delivered by the insulin pump in real time.

**Table 1 pharmaceutics-14-01406-t001:** Insulin analogs, their onset time, time to peak serum concentration, and duration of drug action.

Insulin and Analogs	Time of Onset	Time to Peak Serum Concentration (Cmax)	Duration of Drug Action
Neutral protamine Hagedorn	1–3 h	6–10 h	14–24 h
Regular human insulin	30 min	1 h	5–8 h
Lispro	5 to 15 min	30–60 min	3–4 h
Aspart	10–20 min	40–50 min	3–5 h
Glulisine	20 min	1 h	4 h
Glargine	1–2 h	Peakless profile	24 h
Detemir	1.6 h	Peakless profile	Dosage-dependent; standard dose up to 24 h
Degludec	1 h	Peakless profile	42 h to 4 days
Icodec	N/A	16 h; albumin-bound	1 week

**Table 2 pharmaceutics-14-01406-t002:** Modifications of insulin analogs and their impact on the absorption of insulin.

Analog	Type of Insulin Analog	Modification	Impacts on the Absorption of Insulin
**Lispro**	Bolus	Reversal of the insulin’s B28 (proline) and B29 (lysine)	The designed modifications prevent formation of a dimer/hexamer or self-association, resulting in faster absorption of insulin monomers when injected subcutaneously.
**Aspart**	Bolus	Substitution of B28 proline with aspartic acid	Reduce monomer–monomer interaction. Enhance repulsion between charged aspartic acid and nearby glutamic acid B21, causing rapid insulin hexamer dissociation into monomers.
**Glulisine**	Bolus	Two modifications: Asparagine at position B3 substituted for lysine Lysine at position B29 substituted for glutamic acid.	These modifications change the isoelectric point from 5.5 (native insulin) to 5.1, improving the solubility of insulin after subcutaneous injection. This enhances stable dimers and monomers at pharmaceutical concentrations in zinc-free buffer.
**Glargine**	Basal	B-chain C-terminal extension with two arginine residues A-chain position 21 substitution of glycine for asparagine	The isoelectric point increases to 6.7 to enhance the solubility of insulin. The glycine substitution prevents the deamidation effect of asparagine, causing a more stable insulin aggregation for long-term release.
**Detemir**	Basal	Acylation of myristic acid to lysine at B-chain position 29	Detemir binds to albumin and forms a reversible bond, resulting in slow release and prolonged action.
**Degludec**	Basal	Deletion of B-chain position 30 threonine Conjugation of hexadecenoic diacid—B-chain position 29 lysine via glutamate linker	Degludec establishes an insulin depot via insulin multi-hexamer formation in the subcutaneous layer with highly predictable gradual dissociation, resulting in long-term release and action.
**Icodec**	Basal	C20 fatty diacid-containing side chain Three amino acid substitutions (A14E, B16H, and B25H)	The C20 fatty diacid-containing side chain enforces strong, reversible albumin binding and the gradual release of icodec from albumin. The substituted amino acids result in a slower receptor-mediated clearance, prolonging its half-life.

**Table 3 pharmaceutics-14-01406-t003:** Summary of clinical trials on alternative interventions for the management of diabetes mellitus.

Intervention	Target	Clinical Trial Phase	Purpose of Study	Clinical Trial Number
Automated insulin delivery (AID) system (Insulin lispro)	Type 1 diabetes (adult)	-	To evaluate whether the AID system is able to function as designed	NCT03743285 [155] NCT03367390 NCT03848767
	Type 1 diabetes (adult)	-	To evaluate whether the AID system is able to function as intended with personalized basal insulin rates when basal insulin rates increase	NCT03849612 [155]
	Type 1 diabetes (adult)	-	To determine predictive low glucose suspension (PLGS) feature safety and functionality	NCT03890003 [156]
Microneedle	Type 1 diabetes (children and adolescents)	2, 3	To determine the difference in glycemic control between subcutaneous insulin catheters and microneedles for bolus delivery	NCT00837512 [157]
PaQ Insulin infusion device (Insulin aspart)	Type 2 diabetes (adult)	-	To evaluate the ability of patients to use PaQ (patch on insulin delivery device) for the control of blood glucose	NCT01535612 [158]
PaQ Insulin delivery device (Insulin aspart)	Type 2 diabetes (Adult)	-	To evaluate the efficacy and safety of basal bolus insulin delivery with PaQ in insulin	NCT02419859 [159]
iLet bionic pancreas (insulin aspart)	Type 1 diabetes (adult)	2	To evaluate the safety of insulin aspart in a different insulin delivery setting in the iLet	NCT03816761 [160]
iLet bionic pancreas (BP) (Insulin lispro or aspart)	Type 1 diabetes (children, adult)	-	To compare the insulin-only configuration of the iLet BP system in maintaining normal glycemia compared to usual care in a home-use setting	NCT04200313
Closed-loop control system using JDRF artificial pancreas	Type 1 diabetes (adult)	Early phase 1	To determine real-time continuous glucose sensing with automated insulin delivery in a closed-loop system	NCT01484457
Closed-loop insulin delivery system 670 G	Type 1 diabetes (children, adult)	-	To track the initiation of the FDA-approved 670G closed-loop insulin delivery system	NCT03017482 [161]
Closed-loop insulin device	Type 1 diabetes (children 6 months to 7 years)	-	To evaluate the efficacy of closed-loop insulin pump therapy in delivering insulin in children less than 7 years of age	NCT01421225 [162]
Imperial College closed-loop insulin device (bio-inspired artificial pancreas)	Type 1 diabetes (adult)	-	To evaluate the safety and efficacy of closed-loop insulin pump therapy in delivering insulin in people with type 1 diabetes	NCT01534013 [163]
V-Go device (Humulin and insulin lispro or aspart)	Type 2 diabetes (adult)	-	To determine the efficacy of regular human insulin on the V-Go device as compared to rapid-acting insulin in the V-Go device	NCT03495908 [164]
FMPD (Insulin aspart and glucagon)	Type 1 diabetes (adult)	2	To compare the glycemic control in persons with type 1 diabetes using a fading memory proportional derivate (FMPD) algorithm with insulin plus glucagon vs. the FMPD insulin-alone algorithm	NCT00797823
PassPort (R) transdermal insulin delivery system	Type 1 diabetes (adult)	1, 2	To evaluate the pharmacodynamics and pharmacokinetics of the PassPort(R) transdermal insulin patch	NCT00519623
Medtronic miniMed implantable pump (human recombinant insulin)	Type 1 diabetes (adult)	3	To evaluate the effectiveness of an implantable insulin delivery pump to reduce severe hypoglycemia compared to subcutaneous insulin (MIP310)	NCT00211536

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
