# Peer review of "A Comprehensive Review of the Evolution of Insulin Development and Its Delivery Method"

_pharmaceutics, 2022, doi:10.3390/pharmaceutics14071406_

Round 1

Reviewer 1 Report

The manuscript is suggested to be accepted in current version. 

Author Response

Dear Editor,

The manuscript has been revised according to the suggestions and comments of the reviewers. The major revised parts can be tracked in the word file for your convenience of re-reviewing. The responses to the specific comments of the reviewers are as follows:

Reviewer 1

The manuscript is suggested to be accepted in current version. 

Response to reviewer 1

We would like to thank reviewer 1 for the kind comments.

Reviewer 2 Report

General note: it would be much better if the citations were discussed in 3 sections: in vitro studies, studies using animal models, and finally, clinical studies. 

Line 28: 'chronologies' is wrong grammatically. 

Line 65: replace 'resultant' with 'result'. 

Line 78: need 'and' before 'triggers'.

Line 101: 'to receive multiple times of' is wrong grammatically. At this juncture, I recommend the whole manuscript be edited for English grammar. 

Line 211: correct spelling for 'enhancers'.

Line 231: 'This study' ... is it citation 70 or 71? Vague. 

Line 244: should read 'insulin' not 'inulin'. 

Line 333: 'it was good to note' ... not sure which paper the authors are referring to here. 

Line 470: 'rectal' to be replaced by 'rectum'.

Line 571: mention whether this was a preclinical or clinical study. Do this for all your citations please. 

Lines 608-612: awkward writing. Correct grammatically. 

Line 630: comment on what the authors of [132] stated as reasons for development of lung cancer. 

Line 685: its almost as if [142] was forcibly inserted at the end of that paragraph. Elaborate further on that paper or remove. 

Table 1: is confusing, as some patent numbers are repeated, and some rows are same as the one above/below it. Redo to make clear for the reader. 

Line 708: it is imperative for the authors to make it very clear that none of the new technologies have superseded the trusted sc administration. I would recommend that this statement be put in the Abstract too. 

Line 725: there is no need to use upper case for the research scholarship name. Looks awkward. 

Author Response

Dear Editor,

The manuscript has been revised according to the suggestions and comments of the reviewers. The major revised parts can be tracked in the word file for your convenience of re-reviewing. The responses to the specific comments of the reviewers are as follows:

Reviewer 2

General note: it would be much better if the citations were discussed in 3 sections: in vitro studies, studies using animal models, and finally, clinical studies. 

Line 28: 'chronologies' is wrong grammatically. 

Line 65: replace 'resultant' with 'result'. 

Line 78: need 'and' before 'triggers'.

Line 101: 'to receive multiple times of' is wrong grammatically. At this juncture, I recommend the whole manuscript be edited for English grammar. 

Line 211: correct spelling for 'enhancers'.

Line 231: 'This study' ... is it citation 70 or 71? Vague. 

Line 244: should read 'insulin' not 'inulin'. 

Line 333: 'it was good to note' ... not sure which paper the authors are referring to here. 

Line 470: 'rectal' to be replaced by 'rectum'.

Line 571: mention whether this was a preclinical or clinical study. Do this for all your citations please. 

Lines 608-612: awkward writing. Correct grammatically. 

Line 630: comment on what the authors of [132] stated as reasons for development of lung cancer. 

Line 685: Its almost as if [142] was forcibly inserted at the end of that paragraph. Elaborate further on that paper or remove. 

Table 1: is confusing, as some patent numbers are repeated, and some rows are same as the one above/below it. Redo to make clear for the reader. 

Line 708: it is imperative for the authors to make it very clear that none of the new technologies have superseded the trusted sc administration. I would recommend that this statement be put in the Abstract too. 

Line 725: there is no need to use upper case for the research scholarship name. Looks awkward. 

Response to reviewer 2

Thank you for your constructive comments.

Spelling mistakes pointed out for Line 28, Line 65, Line 78, Line 101, Line 211, Line 244, Line 470 have been corrected.

Line 231, Line 333 has been modified to better represent the referred citations.

Lines 608-612 have been revised on the grammatical errors.

Line 630: in reference to the citation [132]; which has now moved down to citation [150], we have commented on stated reason for the development of lung cancer with the use of Exubera.

Previous citation [142], which has changed to [160], we have elaborated further on the citation.

Table 1 has been re-arranged for a better understanding. We have grouped similar interventions in order to not confuse the readers. Some of the intervention has used similar names but their proposed study or the aim of the study is different from one another.

Line 708: This comment has been added to both the abstract and the conclusion.

We have also discussed and divided each of the delivery routes into in vitro and in vivo subsections.

Reviewer 3 Report

This manuscript systematically review the evolution of insulin development and Its delivery method. This manuscript was well prepared. The logic flow was clear.  I think this manuscript may be interesting to the readers of Pharmaceutics. However, pharmacokinetics of insulin analogues seem unclear. How did insulin analogues were absorbed, distributed, metabolized and excreted in body? If the authors can add more details or mechanisms of insulin analogues pharmacokinetics, that will be much better. 

Author Response

Dear Editor,

The manuscript has been revised according to the suggestions and comments of the reviewers. The major revised parts can be tracked in the word file for your convenience of re-reviewing. The responses to the specific comments of the reviewers are as follows:

Reviewer 3

This manuscript systematically review the evolution of insulin development and Its delivery method. This manuscript was well prepared. The logic flow was clear.  I think this manuscript may be interesting to the readers of Pharmaceutics. However, pharmacokinetics of insulin analogues seem unclear. How did insulin analogues were absorbed, distributed, metabolized and excreted in body? If the authors can add more details or mechanisms of insulin analogues pharmacokinetics, that will be much better. 

Response to reviewer 3

Thank you for the comments.

We fully agree to the suggestion given by reviewer 3. Therefore, we have added a section on the different insulin analogues and their respective pharmacokinetic/pharmacodynamic (PK/PD) profiles as well as potential mechanism.

Round 2

Reviewer 2 Report

Well done.